# Novel Hydrogels Based on the Nano-Emulsion Formulation Process: Development, Rheological Characterization, and Study as a Drug Delivery System

**DOI:** 10.3390/pharmaceutics16060812

**Published:** 2024-06-14

**Authors:** Usama Jamshaid, Nicolas Anton, Mohamed Elhassan, Guillaume Conzatti, Thierry F. Vandamme

**Affiliations:** 1INSERM (French National Institute of Health and Medical Research), UMR 1260, Regenerative Nanomedicine (RNM), FMTS, Université de Strasbourg, F-67000 Strasbourg, France; usamapharmacy2012@gmail.com (U.J.); mohamed-ahmed.mohamed-elhassan-ahmed@etu.unistra.fr (M.E.); conzatti@unistra.fr (G.C.); 2Faculty of Pharmacy, The University of Lahore, Lahore 54590, Pakistan; 3Department of Pharmaceutics, Faculty of Pharmacy, University of Gezira, Wad Medani 21111, Sudan

**Keywords:** nano-emulsion, hydrogel, sol/gel transition, rheology, ternary system

## Abstract

In this study, we present a new type of polymer-free hydrogel made only from nonionic surfactants, oil, and water. Such a system is produced by taking advantage of the physicochemical behavior and interactions between nonionic surfactants and oil and water phases, according to a process close to spontaneous emulsification used in the production of nano-emulsions. Contrary to the classical process of emulsion-based gel formulation, we propose a simple one-step approach. Beyond the originality of the concept, these *nanoemulgels* appear as very promising systems able to encapsulate and deliver various molecules with different solubilities. In the first section, we propose a comprehensive investigation of the gel formation process and its limits through oscillatory rheological characterization, characterization of the sol/gel transitions, and gel strength. The second section is focused on the follow-up of the release of an encapsulated model hydrophilic molecule and on the impact of the rheological gel properties on the release profiles.

## 1. Introduction

Nano-emulsions are a promising alternative to drug permeation as well as targeting poorly soluble drugs by increasing its availability [1]. In addition to the stable dispersion of lipophilic molecules in aqueous media, nano-emulsion formulations have numerous potential applications in drug delivery and targeting for diagnostic purposes. At the same time, an important strength of nano-emulsions is their safety and compatibility with biological applications [2]. A new formulation of nano-emulsion, the so-called *nanoemulgel*, has been developed by the current research, exhibiting the properties of a hydrogel without use of polymers which appears promising to control the delivery of hydrophilic Active Pharmaceutical Ingredients (APIs) in the form of nano-carriers for topical applications and in situ depot systems. Nano-emulsions are kinetically stable liquid-in-liquid dispersions with droplet sizes below 200–300 nm [3]. Nano-emulsions are, in general, optically clear and translucent—in function of the droplet concentration, oil-in-water or water-in-oil dispersion, and are generally stabilized by nonionic surfactants [4,5]. According to the spontaneous emulsification process to generate nano-emulsion dispersions, oil and surfactants are first mixed in specific stirring at appropriate temperature, and then the water phase is added forming the oil-in-water nano-droplets [6]. In case the amount of water is drastically reduced in such a process, the nano-emulsion does not form, but instead, the ternary system adopts the properties of a hydrogel, the so-called *nanoemulgel*. These gels have never been studied before, neither their formation process, rheological characterization, nor their use as drug delivery systems. This ternary system is, by definition, a bi-continuous mixture of oil and water phases with specific rheological properties of hydrogels and is a prime candidate for hybrid drug delivery. In contrast, literature reports have described hybrid hydrogels made with the gelation of emulsions already produced [2,7,8,9,10,11,12,13,14,15] are indeed different from *nanoemulgel* concept because these examples need the addition of polymers to create a networking matrix in the aqueous phase.

Generally, these examples from the literature describe the gelation of emulsions that were already formulated and for which the external aqueous continuous phase was turned into a gel in the second step. For example, in the works of Khurana et al. [12], the authors first formulated a nano-emulsion, and then they added Carbopol 904 to the bulk of water, resulting in a sol/gel transition of the system upon a pH increase with triethanolamine. They adapted a classical process used for the fabrication of topical hydrogels to incorporate lipid nanocarriers. In the works of Lu et al. [14], the authors described a radical polymerization of cellulose-based chains mediated by cellulose acetoacetate also used as nano-emulsion stabilizers. On the whole, all the nano-emulsion-based hydrogels described in the literature involve a sol/gel transition or polymerization of the aqueous phase, after the nano-emulsions are formulated using polymers. This is basically different from the system we propose—made with water, oil, and surfactants—and for which the *nanoemulgels* are formed before the formation of droplets in the process as a much simpler approach than the conventional processes. In *nanoemulgels*, hydrophilic molecules are potentially encapsulated in the water portion and released by diffusion, but, on the other hand, lipophilic ones will be released as nano-emulsion droplets upon further dilution, for example, in release media. Thus, *nanoemulgels* are very different from other lipid carriers like SLNs (solid lipid nanoparticles), NLCs (nanostructured lipid carriers), or liposomes. These nanocarriers are free and very mobile in water where they can diffuse in aqueous media. In contrast, *nanoemulgels* behave as a reservoir of lipid carriers formed with the water, which is in contact with the gel and allows a slow release of the nano-emulsion droplets that are gradually formed. Accordingly, as conventional nanocarriers are compatible with administration in blood, *nanoemulgels* will only be administrated according to the subcutaneous or intramuscular routes.

In addition, when compared with nano-emulsions, the viscoelastic properties of *nanoemulgels* may open compatibility with new administration route and new related applications along with the controlled and sustained drug release. *Nanoemulgels*, as a simple and adaptative polymer-free formulation that incorporates drug reservoirs, are innovative and unique drug delivery systems, in comparison to research reported in recent literature [16,17,18,19]. In general, emulsions are unstable systems from a thermodynamic point of view because of the structures of droplets dispersed in the second phase. However, it is interesting to note that such *nanoemulgels* can be considered thermodynamically stable because they are a homogeneous mixture of oil, water, and surfactants. This system, to a certain extent, can be comparable to microemulsions. It should be noted that we can also find a variety of gels reported in the literature that are similar to *nanoemulgels*, though with some basic differences. Among these gels, we can cite the families of *organogels*; e.g., in the works of Vintiloiu and Leroux [20], the authors described organogels made without water. These are made from compounds that create a network in organic media, like lipophilic derivative molecules like peptides or sugars. These gels are formed in an organic phase with a relative stability that often leads to destabilization with time.

In general, *organogels* are described as being formulated without water, indicating that the lipophilic linkers or polymers can create a strong network in the organic phase. This is precisely the main difference with *nanoemulgels*, which need the presence of water to create the gel. This is also illustrated in other works, such as Murdan et al. [21,22], where this aspect appears recurrent, even with the possible addition of water [22], which modifies the gel properties.

In this study, we propose an in-depth investigation of the physicochemical properties of these *nanoemulgel’s* gelation mechanisms and of the impact of the system composition on the viscoelastic properties of the gel. A systematic rheological characterization was performed upon the gradual modifications of the formulation parameters. The second part of this study attempts to establish a relationship between the rheological results and the release profiles.

## 2. Materials and Methods

### 2.1. Materials

Several oil phases were compared: medium-chain triglycerides (MCTs, Labrafac^®^ WL 1349, from Gattefossé, Saint Priest, France), vitamin E acetate (Tokyo Chemical Industry Co. Ltd., Tokyo, Japan), monoglycerides and diglycerides of medium-chain caprylic acids (Capmul^®^ MCM C8, from Abitec Corporation, Colombus, OH, USA), and castor oil from Sigma-Aldrich (Saint-Quentin-Fallavier, France). The nonionic surfactant we used, Kolliphor^®^ ELP (PEG-35 ricinoleate), was obtained from BASF (Ludwigshafen, Germany). Milli-Q water was obtained from a Millipore filtration system. All chemicals were of analytical grade.

### 2.2. Preparation of Nanoemulgels

The preparation of *nanoemulgels* followed the classical formulation process of spontaneous nano-formations as previously reported [6], with the main difference being the reduced amount of water. The first step required the mixing of oil with nonionic surfactants, heated for 10 min at 70 °C with continuous stirring at 1000 rpm in a thermomixer (Eppendorf, Hamburg, Germany). Water was then suddenly added to this first (oil + surfactant) phase, forming the ternary system, leading to the generation of a gel under a specific composition range. This mixture was stirred at room temperature under vortex until it reached a homogeneous aspect.

In this study, the formulation parameters, i.e., the ratio between oil and surfactant and amounts of water in the system, are expressed according to the surfactant-to-oil weight ratio (SOR) and the surfactant plus oil-to-water weight ratio (SOWR), defined as $SOR=\frac{w_{S}}{w_{S}+w_{O}}\times 100$ and $SOWR=\frac{w_{S}+w_{O}}{w_{S}+w_{O}+w_{W}}\times 100$, where $w_{S}$, $w_{O}$, and $w_{W}$ are the weights of surfactant, oil, and water, respectively.

### 2.3. Characterization of the Formulations

#### 2.3.1. Rheological Characterization

Rheology was performed on a HAAKE MARS Rheometer (Thermo Scientific, Waltham, MA, USA) using oscillatory mode with a plane/plane geometry and a gap of 0.1 mm, with the plate rotor (P35/Ti/SB) and sample hood each having a diameter of 35 mm. Amplitude sweep experiments were performed with a frequency of 1 Hz. The storage modulus (G′), the loss modulus (G″), and the phase angle (*ϕ*) were extracted within the linear viscoelastic region (strain = 1%). Here, amplitude sweep experiments allowed determining the linear domain, and we chose to use the values of G′ and G″ at a fixed frequency of 1 Hz to observe a comparative study between several formulations at a fixed frequency within the linear regime. As the moduli are dependent on the frequency of deformation, a frequency of 1 Hz was chosen as an intermediate time of motion to be representative of the systems.

#### 2.3.2. Characterization of Nanoparticle Size Distribution

The particle size distribution, mean size, and polydispersity indexes (PDIs) were determined by dynamic light scattering (DLS) using a Malvern^®^ Nano ZS instrument (Malvern, Orsay, France). DLS data were analyzed using a cumulant-based method. Size distribution and polydispersity index (PDI) were recorded at a temperature of 25 °C after controlling dilution 10,000 times. The experiments were performed in triplicate.

#### 2.3.3. In Vitro Release Studies of Methylene Blue

To evaluate the ability of the *nanoemulgels* to encapsulate and release hydrophilic molecules, the aqueous phase was loaded with a hydrophilic dye (methylene blue, MB, 7 mg/mL). After formulation, 1 g of the MB-loaded *nanoemulgels* was taken and placed in a small crystallizer and then immersed in a beaker filled with MilliQ water (800 mL) under gentle magnetic stirring (100 rpm). Aliquots of release media were then collected at different time intervals until complete MB release was reached. The time intervals were 2.5 min within the initial burst and then every 10 min, up to 2 h. Evaluations were performed by visible spectroscopy (Thermo Scientific Varioskan LUX Multimode, Waltham, MA, USA) at the λmax of 668 nm.

### 2.4. Statistical Analysis

Quantitative data were expressed as the mean ± standard deviation obtained from three independent experiments.

## 3. Results and Discussion

### 3.1. Oscillatory Rheological Characterization

The first series of experiments relates to the characterization of ternary systems using classic nano-emulsion ingredients, i.e., MCT, Kolliphor^®^ ELP, and water. A SOR of 60% was first fixed, and the SOWR was varied from 40% to 80%. Representative rheograms showing G′, G″, and phase angle *ϕ* before the sol/gel transition (SOWR = 44%), at the gel point (SOWR = 52%), and at the gel state (SOWR = 60%) are shown in Figure 1. The details of the composition are reported in Appendix A. G′, G″, and phase angle *ϕ* are represented as functions of the strain. This representation is generally proposed to show the linear regime clearly visible from around 0.1% to 5%.

In the linear regime, a phase angle from 45° to 90° indicates a prevalence of the viscous contribution G″ upon the elastic modulus G′ (Figure 1a,d). During the sol/gel transition, elastic and viscous moduli show similar values, and the phase angle *ϕ* presents a transitional value between a pure viscous state and a pure elastic state of around 50° (Figure 1b,e). Finally, when the gel was formed (Figure 1c,f), the elastic contribution G′ prevailed upon G″, and the phase angle dropped to a value close to 0°, indicating a purely elastic system. This first result demonstrated that the water content had a significant role in the viscoelastic behavior of the formulation. In general, nano-emulsion formulations are performed using SOWRs ranging from 20 to 30%, i.e., with a much higher water content. These results showed that the ternary system could be considered liquid up to a SOWR ~50% and transit to a gel state when lower amounts of water are involved in the nano-formulation.

Other sets of experiments were performed by increasing the proportion of surfactants in the mixture, using SORs = 40%, 60%, and 70%, and extended to other oils. Thus, *nanoemulgels* were formulated with monoglycerides (Capmul^®^), vitamin E acetate, and castor oil in the same conditions as described with Labrafac^®^. Their respective results appear in Figure 2 (for G′) and Figure 3 (for the phase angle). The first observation is about the strength of the gels, in a comparable range > 10 MPa for Labrafac^®^, castor oil, and vitamin E, when it is significantly weaker for Capmul^®^. For this reason, the sol/gel transition is clearer when studying Labrafac^®^, castor oil, and vitamin E acetate, as we clearly observed a dramatic drop in *ϕ* to be close to zero, at low water content (SOWR = 40–50%). However, when the water amount was further decreased, the phase angle tended to rise again towards viscous values, making the gel weaker. The system tends to be a pure surfactant/oil phase, emphasizing the important role of water in the formation of the gel state. For most of the formulations, we observed a plateau that corresponded to the gel state of the formulation. Higher amounts of surfactants seem to shift the gel formation region towards lower SOWRs, i.e., toward a higher amount of water. On the other hand, the lower amount of surfactant (SOR = 40%) gave rise to significantly lower moduli and gel strength, compared to higher SOR = 60% and 70%. The stronger gel was obtained with Labrafac^®^ and castor oil, giving $G^{’}>10^{4}$ Pa (at SORs = 70% and 60% for SOWR = 50–70%); vitamin E acetate also exhibits a similar strength in the range with a dependence on the water amount, i.e., SOWR = 40–55% for SOR = 70% and SOWR = 50–65% for SOR = 60%.

As a last observation, using a monoglyceride as the oil phase—Capmul^®^—shows a behavior that was significantly different when compared with the other three oils, generating a weak gel with a maximum of $G^{’}=\left(230\pm 20\right)$ Pa (SOR = 70% and SOWR = 62%), along with a changed transition behavior. This different behavior, compared with other oils, can be attributed to the fact that Capmul^®^ MCM C8 is less hydrophobic and presents a structure close to a lipophilic surfactant, even having an HLB~4.7 [23]. As a result, it can lead to a less stable water/oil interface and a weaker gel. Herein, it can be seen that if the SOWR is further decreased to below 40%, the water amount becomes too high and the ternary system becomes a classical emulsion, as seen with the other oils.

A correlation between Figure 2 and Figure 3 can be achieved; the phase angle variations clearly follow the rise of the storage modulus *G*′, confirming the formation of a hydrogel. Hence, the gel strength appears to be strongly dependent not only on the nature of the oil but also on the SOR and SOWR. A general trend suggested that the higher the surfactant amount, the stronger the gel strength. A second observation was that as the surfactant amount increased, the sol/gel transition was shifted towards a lower SOWR, i.e., *nanoemulgels* were formed with more water. However, these observations were not followed with Capmul^®^ oil.

Interestingly, a comparison with classical hydrogels gives the *nanoemulgels* between 3% agarose hydrogels [24] and 2% calcium reticulated alginate [25], giving *G*′ around 100 kPa and below 10 kPa, respectively. The visual and digital evaluations of these gels evidenced thick and solid-like gels for all *nanoemulgels* described, except for Capmul^®^ systems, which behave as a soft gel, in accordance with their lower measured modulus.

To summarize, the rheological characterization emphasizes the formation of a gel that is relatively strong and spontaneously generated when mixing water/oil/surfactants according to a specific protocol and order. Without water, the oil/surfactant system is an oily liquid. Upon the addition of water, and for the lower amounts of water (high SOWR), the ternary system is still a liquid, gradually going to the sol/gel transition. Increasing the water concentration results in the formation of a strong hybrid gel where aqueous and lipid phases are blocked together and their interface stabilized with surfactants. This ternary system probably behaves as a gel because of the structuration of the aqueous and oil domains, in which interfaces mechanically interact with each other on small scales with a high specific surface. In this hypothesis, the larger the interfacial area—due to a larger surfactant amount—the stronger the gel. Finally, the further addition of water results in the gel’s dislocation, along with the formation and release of lipid nano-emulsion droplets. The impact of increasing the surfactant amount appears to be increasing the gel’s strength and shifting the gelation domain to a lower SOWR. It might be due to the higher capability of the entrap water phase due to the higher water/oil interfacial area. When SOR = 40%, the low amount of surfactant may not permit an efficient surfactant/water-binding network to form, explaining why the gels formed appear less strong in comparable water content. The last point arising from this series of results is the influence of the nature of oil that likely modifies the interactions between the oil phase and surfactant, impacting the surfactant’s efficiency.

### 3.2. Polydispersity Index and Size Analysis of the Nano-Emulsion-Based Systems

When *nanoemulgels* are maintained in contact with the aqueous phase, nano-emulsions are produced according to the spontaneous nano-emulsification process, exactly when water is immediately added.

The results are reported in Figure 4, which demonstrate, as is commonly known, that the higher the quantity of surfactants, the smaller the particle size and PDI. The water amount in the gel, SOWR, does not significantly impact the nano-emulsion particle size, giving, with Labrafac^®^, the following mean values: $d=159\left(22\right)$ nm, $65\left(28\right)$ nm, and $29\left(3\right)$ nm (according to the conventional notation *d* = mean (standard deviation)), for SORs of 40%, 60%, and 70%, respectively. The PDI follows the same trend, showing better monodispersity for larger surfactant amounts—a classical behavior of spontaneous emulsification [6]. In addition, the nano-emulsion size obtained by this two-step process, i.e., (i) the fabrication of *nanoemulgels* and (ii) their dissolution, is similar to those obtained by using a large water amount, for instance SOWR = 20%, generating nano-emulsions in a single step [6]. Along with the fact that the SOWR does not impact the particle size, it also signifies that we can consider *nanoemulgels* an intermediate stage in nano-emulsion formation.

A comparison of nano-emulsion size distribution with the different oils used above is compared with a representative formulation (SOR = 60% and SOWR = 50%), giving hydrodynamic radii of 44 nm (PDI = 0.10), 135 nm (PDI = 1.00), 27 nm (PDI = 0.17), and 34 nm (PDI = 0.10), for Labrafac^®^, Capmul^®^, castor oil, and vitamin E acetate, respectively. Labrafac^®^, castor oil, and vitamin E acetate also showed very small particle sizes in a close range, whereas Capmul^®^ emulsions revealed a rough polydisperse dispersion that cannot be considered a nano-emulsion as its PDI is equal to one. The absence of nanostructuration can explain the absence of strong gel formation. Hence, it is clear that nanoemulgel formation is directly related to its ability to arrange in a nanoscale architecture.

### 3.3. Release of Hydrophilic Model Dye from Nanoemulgels

In the previous section, we described the formation and rheological properties of *nanoemulgels* based on the formulation parameters. In this section, we propose to evaluate the influence of the formulation parameters, i.e., the influence of the gel properties, on the release of a model dye encapsulated in aqueous phase. Here, we chose to study a model molecule to understand the behavior of the drug delivery system and the impact of the formulation parameters on the release of these molecules. In comparison with marketed products with the same delivery profiles, e.g., subcutaneous form with long release profile, implants, microspheres, etc., the strengths of nanoemulgels are the simplicity of the formulation and the simplicity of the composition, without polymers or without the use of organic solvents in the formulation process.

Figure 5 presents the MB release over time from Labrafac^®^-based formulations. As expected, the dye was rapidly released in the external medium through direct diffusion from the water phase of the emulsion (release time < 30 min). However, in some conditions, we clearly observed a slowdown of the release profile, with a 100% release significantly delayed for SOWR = 70% and SOR = 60% and 70%. In general, the release properties of a molecule are related to its properties, Mw, solubility, pKa, and environmental conditions. However, in the present study, as the release molecule (MB) is the same over all the studies, we assumed that the different release profiles were fully comparable since the gels have been formulated with the same compositions. In general, the slowest release profiles correspond to a lower amount of aqueous phase. If we assume that these *nanoemulgels* are formed with aqueous and oil domains mixed as a bicontinuous phase in a submicron range, these profiles show the kinetics for the dye to escape such a network without speculating on the release mechanism. The formulations for which the SOWR = 70% are the ones with minimized aqueous phases and thus with the smallest water domains compared to oil ones, which possibly explains the extended time necessary for the dyes to escape the gel compared with the lower SOWR. On the other hand, slower release profiles also appear in intermediate SOWR values, e.g., for SOR = 40%/SOWR = 60% and SOR = 60%/SOWR = 50% in Figure 5a,b, values of SOWR that seem to correspond to the sol/gel transition, as can be seen in Figure 2a. Considering the structure of the gels, where the aqueous and oil domains are mixed in a submicron-scale range, the behavior of the gels is changing at higher water amounts, which might be due to the smallest scale of the gel structuration. Increasing in the water proportion may enlarge the water channels and water network, making it easier for the leakage of encapsulated hydrophilic molecules. Additionally, the fact that for the SOR = 40% and SOWRs = 60% and 70%, a prolonged release behavior was observed for the solute molecule, despite the absence of the formation of a strong gel, evidenced the decorrelation between the thickening behavior and the release kinetics, likely due to some other considerations like the bi-phasic organization of water and oil.

Release profiles with SOR = 70% demonstrated delayed behavior as compared with formulations with fewer surfactants, but they gradually slowed down with decreasing water amounts. In order to understand the release mechanisms, these release profiles are analyzed with the Korsmeyer–Peppas model in our case of unidirectional diffusion of releasing compounds:$$\frac{M_{t}}{M_{\infty }}=k\cdot t^{\alpha }$$
where $M_{t}/M_{\infty }$ is the fraction of dye released at the time t; k is the release rate; and α is the release exponent. In our case, the release exponent is an indication of the type of diffusion: Fickian when α≤0.5 and non-Fickian for 0.5<α<1. Values of the release exponent are reported in Figure 5d, showing the notable threshold at α=0.5 with a dotted line. The mean values and standard deviations of the release exponent α are obtained by a statistical analysis of the experimental data. As the experimental data can be considered as a set of N samples (x_i_, y_i_), with individual errors on the measurements y_i_, and considering the fitting power law of the form $y=k\cdot x^{\alpha }$, the mean and standard deviation of the parameters k and α, corresponding to the best-fitting power law, are estimated using a simple statistical analysis performed with MATLAB (R2024a) based on Monte Carlo sampling. These results emphasize an increase in α, with both the SOR and SOWR inducing a transition between Fickian and non-Fickian regimes. Basically, the Fickian regime can be associated with the direct escape of the dyes from a communicating and porous gel network with straight pathways, while the release profiles are non-Fickian when the molecules need to escape from a convoluted pattern where molecules travel a longer distance—compared to Fickian regime—before escaping the gel network. According to this hypothesis, at a constant surfactant amount, for a SOR = 40% or SOR = 60%, the increase in α with the corresponding decrease in water amount results in reducing the connection between the water veins, thus increasing the distance for the dye to escape. It appears that at higher surfactant amounts (SOR = 70%), or when increasing the surfactant amount (SOR = 40–60%), the water–oil interfacial area is higher, the scale range of the morphology is reduced, and the samples are more rapid in the non-Fickian regime. The more complex gel morphology might exist at the sol/gel transition, for which the release exponents reach a maximum, as raised above. Herein, we show that *nanoemulgels* can release hydrophilic molecules. In addition, we can also assume that these gels can release lipophilic molecules through the release of nano-emulsion droplets. As shown in Figure 4, *nanoemulgels* released nano-emulsions as one droplet population with a narrow monodispersity. On the other hand, it is well acknowledged in the literature that nano-emulsions are efficient and stable carriers of lipophilic molecules [26,27], thus making *nanoemulgels* a potential system to also release lipophilic molecules.

### 3.4. Impact of the Nature of the Oil and Water Phases on the Release Profiles

The impact of several other conditions on the release profile, changing oil nature or the addition of a thickener in water, was also evaluated. To this end, representative *nanoemulgel* formulations were selected (SOR = 60%/SOWR = 50%), and MB release profiles were determined with different oils (Labrafac^®^, Capmul^®^, castor oil, and vitamin E acetate) with and without a thickener in the aqueous phase. The idea was to first compare the impact of the oil nature of the profiles, as performed in the rheological characterization and, secondly, to emphasize whether the modification of the properties of the aqueous phase can potentially delay the dye release in the gel network. The results are reported in Figure 6, showing slight differences in the release profiles between the different oils (Figure 6a) and not necessarily correlated with the strength of the gel or the sol/gel state. All of the formulations presented a slight release retention, but castor oil revealed a relatively slower release, with a time of 50% between 20 and 30 min but a time of 100% around 60 min, while all other formulations released 90% or more of their dye within 20 min. It is important to note that this experimental setup can be considered the worst case that can lead to faster release kinetics, and, in other conditions, e.g., subcutaneous, with less aqueous drainage, the release kinetics could likely be slower. From this observation, we can conclude that the morphology of the formed gel also depended on the nature of the oil phase, probably impacting the morphology and size of the bicontinuous oil/water network. Adding hydroxylethylcellulose (HEC) to water results in significant modifications of the profiles (Figure 6b). HEC exhibited a basic hydrophilic moiety that can help it adsorb onto the interface. As a main observation, the presence of HEC in water reduces the burst release of MB, either due to the slowing of MB diffusion in aqueous channels or modifying the gel morphology. Notably, the addition of HEC to castor oil *nanoemulgels* did not significantly modify the release profile of this formulation, with a total release after the same period (60 min). Its presence in the water phase does not modify the gel structure and suggests that HEC does not directly interact with the dye. The differences observed for the other formulations can be more reasonably attributed to changes in the gel morphology.

## 4. Conclusions

In this study, we present a new type of material intended for the delivery of hydrophilic and lipophilic drugs in the form of a viscoelastic gel matrix, formulated without polymers. This hydrogel, the so-called *nanoemulgel*, is formulated as an intermediate stage during the formulation of nano-emulsions by spontaneous emulsification and releases nano-emulsion droplets when it is diluted. In this study, rheological characterization identified a specific range in terms of the amount of water where the gel is formed when, for low or high water content, the system is liquid. The impacts of the gel properties of the surfactant amount, water amount, and nature of the oil were investigated and linked to the release properties of a model hydrophilic molecule. Regarding the potential applicability of *nanoemulgels* and the industrial scale-up of the formulation process, as it is a simple mixture of water within the oil and surfactant homogeneous phase, the scale-up challenge lies in the control of the mixture stage, as well as sterilization, that can be performed according to the classical approaches like fabrication in a sterile vessel. Release studies revealed that the gel formation, related to the nano-structuration of the oil/water/surfactant structuration, slows down the leakage of a hydrophilic dye without a clear correlation with the thickening phenomenon. We showed that *nanoemulgels* are interesting tools to obtain a thick formulation with a simple and safe composition, able to administer potential drugs over time.

## Figures and Tables

**Figure 1 pharmaceutics-16-00812-f001:**
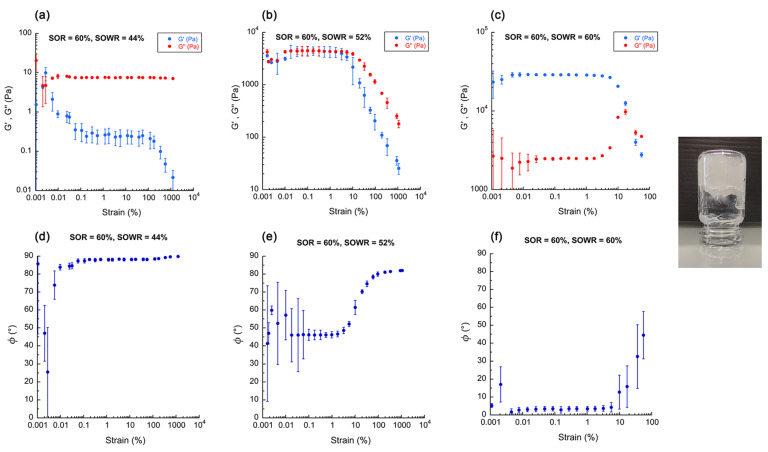
Representative examples of rheograms chosen to show the sol/gel transition for a ternary system composed of MCT (Labrafac^®^ WL 1349), Kolliphor^®^ ELP, and water, at fixed SOR = 60%, and for (**a**) SOWR = 44% (liquid state), (**b**) SOWR = 52% (transition state), and (**c**) SOWR = 60% (gel state, illustrated with the corresponding picture). Experiments were conducted at a constant frequency of 1 Hz. (**d**–**f**) represent the respective phase angles as function of the strain. The data are presented as mean values ± standard deviations; n = 3.

**Figure 2 pharmaceutics-16-00812-f002:**
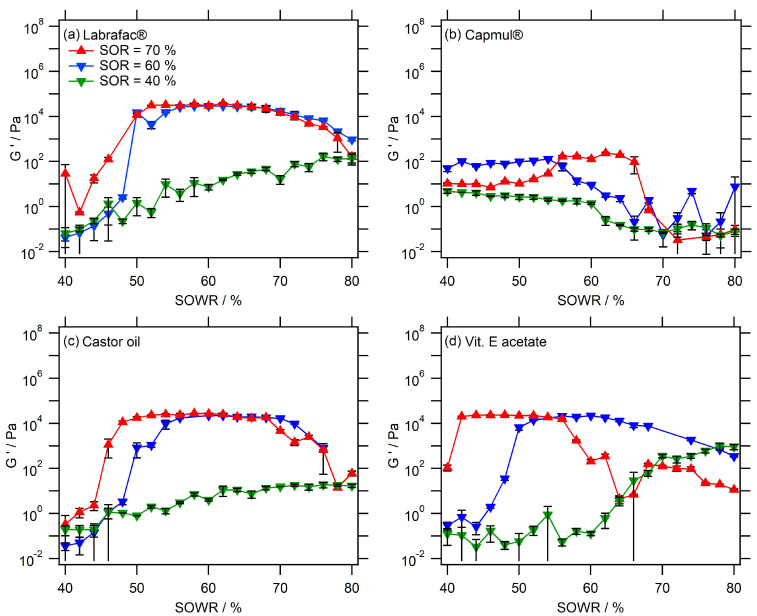
Values of G′ in *nanoemulgels* as a function of the surfactant amount (SOR), water amount (SOWR), and the nature of the oil. *Fixed parameters:* strain = 1% and frequency = 1 Hz. Data are presented as mean values ± standard deviations; n = 3.

**Figure 3 pharmaceutics-16-00812-f003:**
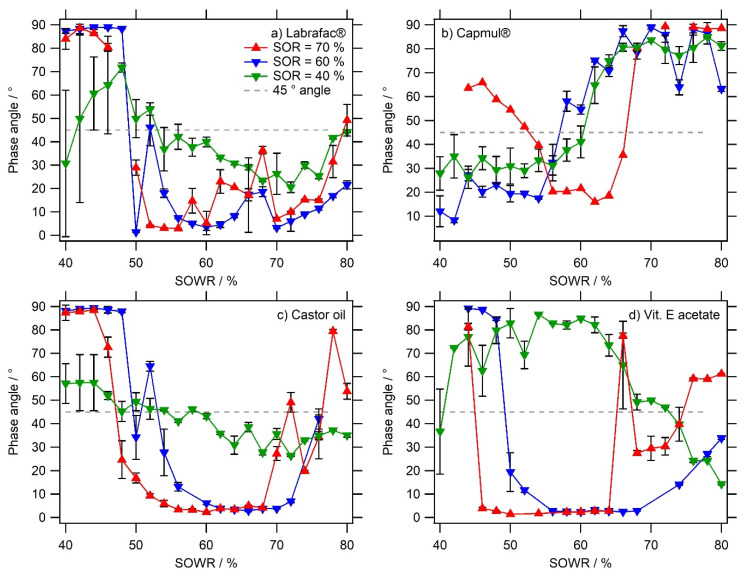
Values of the phase angle *ϕ* in *nanoemulgels* as a function of the water amount (SOWR) and the nature of the oil. The sol/gel transition are indicated with the grey dashed line with *ϕ* = 45°. *Fixed parameters:* strain = 1% and frequency = 1 Hz. The data are presented as mean values ± standard deviations; n = 3.

**Figure 4 pharmaceutics-16-00812-f004:**
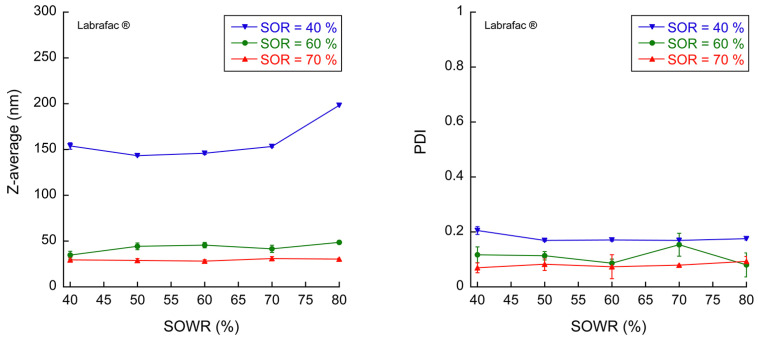
Size and polydispersity indexes of nano-emulsions produced by the dilution of *nanoemulgels* in water as a function of the surfactant amount (SOR) and water amount (SOWR). The oil we used was Labrafac^®^. The data are presented as mean values ± standard deviations; n = 3.

**Figure 5 pharmaceutics-16-00812-f005:**
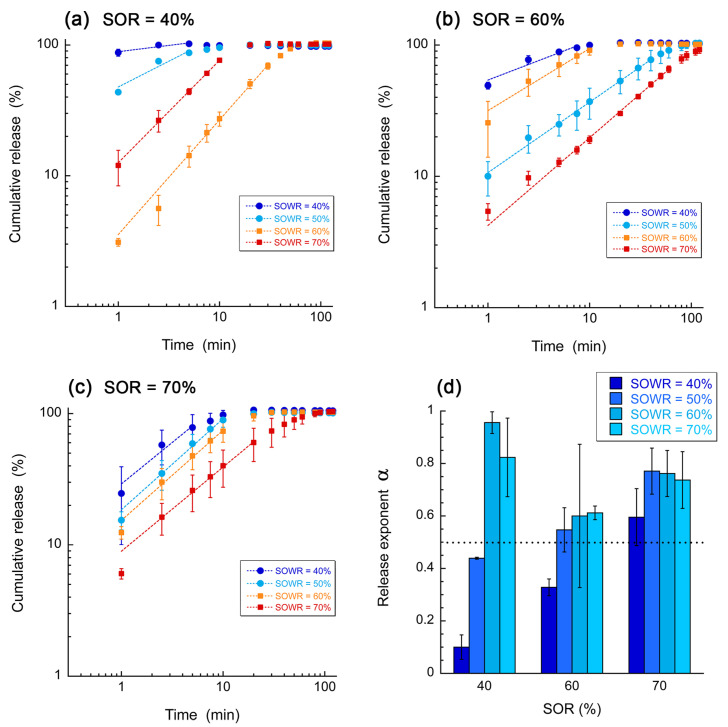
(**a**–**c**) The release profiles of MB in the *nanoemulgels* (oil is Labrafac^®^) for different amounts of surfactant (values of SOR) and the amount of water (values of SWOR). Extrapolation of the data (**a**–**c**) was carried out with the Korsmeyer–Peppas model, and the corresponding values of the release exponent *n* was reported as a function of the different formulation parameters (**d**). The data are presented as mean values ± standard deviations; n = 3.

**Figure 6 pharmaceutics-16-00812-f006:**
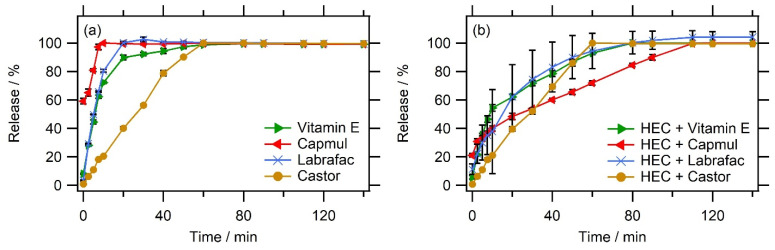
The release profiles of MB in the nanoemulgels (SOR = 60%/SOWR = 50%) for different oils without (**a**) and with (**b**) hydroxylethylcellulose (HEC) as a thickener in the aqueous phase. The data are presented as mean values ± standard deviations; n = 3.

## Data Availability

Data available on demand.

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
