# Peer review of "Novel Hydrogels Based on the Nano-Emulsion Formulation Process: Development, Rheological Characterization, and Study as a Drug Delivery System"

_pharmaceutics, 2024, doi:10.3390/pharmaceutics16060812_

Round 1
Reviewer 1 Report
Comments and Suggestions for Authors
Dear Authors,
I found the manuscript "Novel Hydrogels Based on Nano-Emulsion Formulation Process: Development, Rheological Characterization, and Study as Drug Delivery System" interesting, but I have several comments:
- The generic or chemical name of the nonionic surfactant Kolliphor® ELP should be provided. Line 109
- What kind of water was used: deionized or purified? Should be added. Line 117
- I recommend adding the compositions of the experimental nanoemulgels.
- I recommend adding SOR and SOWR values. Line 123
- Why was a frequency of 1 Hz chosen for the rheological study? A justification should be added. Line 131
- Should be 2.3.3. In Vitro Release Studies of Methylene Blue. Line 142
- Time intervals should be specified. Line 148
- Was the statistical analysis of the data applied?
- I recommend replacing markers (circles, triangles) in graphs with colored lines without markers (Figures 1-3).
- You write "... SOWR was varied from 40% to 80%". However, according to Figure 1, SOWR is 44%, 52%, and 60%. Where are the results for 40% and 80% SOWR? Line 155
- You write "... using SOR = 40% and 70% ...". But it should be SOR = 40–70%. Line 178
- I recommend using particle size instead of droplet size.
- The standard deviation (SD) must be in parentheses (mean (SD)) because the standard error (SE) is given as ± (mean ± SE). I propose correcting it.
Reviewer 2 Report
Comments and Suggestions for Authors
1) Authors should complete the information about authors affiliations, in point b.
2) I think that in Methods, section 2.1. authors should include more detailed information about the components of nanoemulgels, which could be presented in table – the type of ingredients and their amounts. Formulation parameters such as SOR and SOWR are not clear and not easy to understand for the reader.
3) What is the purpose of the release study? How can the date from the conducted study with model dye correspond to the prediction of active substances release? Drug release from emulsions/gel formulations depends on many features e.g. logP, solubility, molecular weight, pKa, etc.
4) The authors did not perform any statistical analysis. Please complete this section.
5) The manuscript contains limited number of studies. I propose to perform other studies such as:
- pH measurements of nanoemulgels
- mechanical and adhesion properties of nanoemulgels
- visual assessement, morphological analysis
- stability studies during storage under various conditions
6) It would be an added benefit if authors presented pictures of sample nanoemulgels.

Round 2
Reviewer 2 Report
Comments and Suggestions for Authors
The manuscript in the present form can be published.